# Resection of Colorectal Liver Metastases with Major Vessel Involvement

**DOI:** 10.3390/cancers16030571

**Published:** 2024-01-29

**Authors:** Janine Baumgart, Sebastian Hiller, Kristina Stroh, Michael Kloth, Hauke Lang

**Affiliations:** 1Department of General, Visceral and Transplantation Surgery, Universitätsmedizin Mainz, 55131 Mainz, Germany; janine.baumgart@unimedizin-mainz.de (J.B.); sebastian.hiller@unimedizin-mainz.de (S.H.); 2Department of Diagnostic and Interventional Radiology, Universitätsmedizin Mainz, 55131 Mainz, Germany; kristina.stroh@unimedizin-mainz.de; 3Department of Pathology, Universitätsmedizin Mainz, 55131 Mainz, Germany; michael.kloth@unimedizin-mainz.de

**Keywords:** colorectal liver metastases, liver surgery, major vessel involvement

## Abstract

**Simple Summary:**

Liver resection for colorectal liver metastases (CRLM) with major vessel involvement is challenging and valid data on outcomes are still lacking. We analyzed data of 32 hepatectomies combined with 35 major hepatic vessel resections and reconstructions with special regard to surgical approaches, histopathological findings and outcome. The vena cava inferior was resected and reconstructed in 19, the portal vein in 6 and a hepatic vein in 10 cases. Histology confirmed a vascular infiltration in 6/32 patients. In conclusion, liver resections with vascular resection/reconstruction are rare, but can be performed with low morbidity and mortality and histological vessel infiltration occurs seldom.

**Abstract:**

Background: Treatment of CRLM with major vessel involvement is still challenging and valid data on outcomes are still rare. We analyzed our experience of hepatectomies with resection and reconstruction of major hepatic vessels with regard to operative and perioperative details, histopathological findings and oncological outcome. Methods: Data of 32 hepatectomies with major hepatic vessel resections and reconstructions were included. Results were correlated with perioperative and oncological outcome. Results: Out of 1236 surgical resections due to CRLM, we performed 35 major hepatic vessel resections and reconstructions in 32 cases (2.6%) during the study period from January 2008 to March 2023. The vena cava inferior (VCI) was resected and reconstructed in 19, the portal vein (PV) in 6 and a hepatic vein (HV) in 10 cases. Histopathological examination confirmed a vascular infiltration in 6/32 patients (VCI 3/17, HV 2/10 and PV 1/6). There were 27 R0 and 5 R1 resections. All R1 situations affected the parenchymal margin. Vascular wall margins were R0. Ninety-day mortality was 0. The median overall survival (OS) for the patient group with vascular infiltration (V1) was 21 months and for the V0 group 33.3 months. Conclusion: Liver resections with vascular resection and reconstruction are rare and histological vessel infiltration occurs seldom. In cases with presumed vascular wall infiltration, liver resection combined with major vessel resection and reconstruction can be performed with low morbidity and mortality. We prefer a parenchymal sparing liver resection with vascular resection and reconstruction to achieve negative resection margins, but in technically difficult cases with higher risk for postoperative complications, tumor detachment from vessels without resection is a most reasonable surgical alternative.

## 1. Introduction

Liver resection remains the main component in the multimodal treatment of patients with colorectal liver metastases (CRLM) offering the best long-term survival rates [1,2,3]. In recent decades, continuous developments and innovations in operative techniques and perioperative management have led to remarkable improvements in safety and outcome of liver resections, increasing the number of patients being eligible for liver surgery. Especially, the introduction of neoadjuvant chemotherapy regimens and downsizing protocols nowadays allows curative treatment approaches even in patients with high tumor burden or multiple liver metastases [4,5,6,7]. Furthermore, the implementation of two-stage procedures with or without portal vein ligation (PVL) or embolization (PVE) to induce a hypertrophy of the future liver remnant (FLR) and to avoid postoperative live failure broadened the spectrum of curative therapy strategies [4,8,9]. The armamentarium was extended by the introduction of a new surgical technique, called Associating Liver Partition and Portal vein Ligation for Staged hepatectomy technique (ALPPS procedure) in 2012 which enables rapid and high hypertrophy rates. In 2015, the interventional technique of liver venous deprivation (LVD) in combination with simultaneous PVE was also implemented with comparable rates of liver regeneration as the ALPPS procedure [8,10].

But, surgical treatment of colorectal liver metastases, which appear to involve major hepatic vascular structures, most notably the portal vein (PV), the vena cava inferior (VCI) or the major hepatic veins (HV) still remains challenging. One main problem in these cases is the fact that neither pre- nor intraoperatively can it be clarified whether the metastases infiltrate or just touch the vascular wall. Hence, in these situations often complex vascular resections and reconstructions or even extensive hepatectomies, which result in a great loss of healthy liver parenchyma, are performed to ensure a R0 resection. These surgical approaches are associated with higher rates of morbidity and mortality. However, current data suggest that an infiltration of the vessel wall by CRLM seems to be a rare setting. Therefore, R1 vascular resection strategies with tumor detachment from major vessels without resection of the vascular wall have become a focus of attention [11,12,13,14]. Nevertheless, valid data are still lacking, and the oncological benefit of the different surgical approaches remains unclear.

The aim of this study was to analyze our institutional experience of hepatectomies in combination with resection and reconstruction of major hepatic vessels due to colorectal liver metastases with special regard to the extent of liver surgery, perioperative morbidity and mortality and outcome related to histopathological proven infiltration of vascular structures.

## 2. Patients and Methods

### 2.1. Patients and Clinical Work-Up

Data of all patients undergoing liver surgery at our high-volume hepatobiliary department are registered in a prospective institutional database. From January 2008 to March 2023, we performed 1236 liver resections for colorectal liver metastases. Data of combined hepatectomies with major hepatic vessel resections and reconstructions were further analyzed for this study. The full work-up included patient’s demographics, perioperative details, histological findings as well as the oncological history and outcome. All patients signed an informed consent for an anonymous analysis of the collected data.

Preoperative diagnostic work-up consisted of clinical examination, elevation of laboratory findings and a contrasted-enhanced computed tomography scan (CT scan) or magnetic resonance imaging (MRI). Additionally, on special request of the surgeon in some cases computer-assisted 3D-reconstructions and 3D-prints (cella®, Murcia, Spain) were performed for better operational planning (Figure 1). 

The graduation of the extend of liver resection based on the Brisbane classification and was adapted for further investigation in minor (<3 segments), major (≥3 segments and right or left hepatectomy) and extended liver resections (extended right or left hepatectomy) [15]. Morbidity was analyzed according to the Dindo/Clavien classification [16] and Vauthey’s criteria defined postoperative liver dysfunction [17]. Mortality was analyzed as in-hospital and 90-day mortality.

Postoperative follow-up included a CT scan or MRI every 3 months during the first year and then every 6 months for another period of 4 years. Follow-up was determined in May 2023.

### 2.2. Hepatectomy Techniques and Vascular Reconstructions

At surgery intraoperative ultrasound of the liver and comparison with preoperative imaging or 3D-prints were performed routinely to ensure reliable detection of CRLM. Surgery aimed at complete removal of all metastases with achievement of a tumor-free resection margin (R0 resection). Both anatomic and non-anatomic resections were performed depending on the extent and localization of liver metastases. When vascular involvement was suggested, the vessel was controlled by placing vascular clamps and then the affected vascular segment was resected. Depending on the extent of the vascular defect, reconstruction was performed by primary suture or by using a peritoneal or bovine patch plastic (vascu-guard synovis^®^) or by interpositioning of a synthetic graft (Gore vascular graft^®^). 

### 2.3. Statistics

Recurrence-free and overall survival was defined as the period from the first stage of surgery until the date of recurrence or death.

Categorical data were compared using χ^2^-test, continuous data of normally distributed data by the Fisher exact test. Survival analyses were performed using the Kaplan–Meier method, and the log-rank test was used to compare median survivals between groups. *p* values < 0.05 were considered significant. 

## 3. Results

### 3.1. Patients’ Demographics and Procedures

Out of 1236 surgical approaches due to CRLM, we performed 35 major hepatic vessel resections and reconstructions in 32 cases (2.6%) during the study period from January 2008 to March 2023. The VCI was resected in 19 cases and was reconstructed by direct suture in 11 and by a bovine patch in 6 cases as well as by a peritoneal patch or gore vascular graft in 1 case, each. The portal vein was resected in six patients and was reconstructed by direct suture in all cases. Hepatic veins were involved in 10 cases and were reconstructed as follows: direct suture in 6 and peritoneal patch in 3 cases and bovine patch in 1 case. Types of vascular reconstruction are shown in Table 1. 

The extend of liver resections in these 32 cases was: minor liver resections *n* = 14 (44%), major liver resections *n* = 13 (41%) and extended liver resections *n* = 5 (17%) including 1 ALPPS procedure (Associating Liver Partition and Portal Vein Ligation for Staged Hepatectomy). 

### 3.2. Histopathological Results of Vascular Infiltration

Histopathological examination of the specimens confirmed an infiltration of the vascular structures in 6 of 32 cases (18.75%). In detail, the VCI was affected in 3/19 (15.8%), the HV in 2/10 (20%) and the portal vein in 1/6 (16.6%) cases (Figure 2).

### 3.3. Margin Status

R0-resection margins were achieved in 27 (84.4%) and R1-resection margins in 5 (15.6%) specimens. All R1 situations were detected at the parenchymal margin side. 

### 3.4. Analysis of the Study Population Depending on the Histopathological Result of Vascular Infiltration

Dividing the study population into two groups based on positive or negative vascular infiltration, no significant differences regarding the parameters gender, age, primary tumour site and nodal status were observed. The number of liver metastases, the extent of liver disease and the preoperative administration of systemic therapy also did not differ. Synchronous liver disease occurred more frequently in patients with no vascular infiltration (*p* = 0.04). Patients’ demographics are described more precisely in Table 2. 

There was no significant difference in the extent of hepatectomy or kind of resected vessels between both groups. 

Perioperative morbidity did not differ significantly. The most common complications with a need of intervention were liver abscess (*n* = 3), bilioma, pleural effusion and pneumonia (*n* = 2, each). In-hospital mortality and 90-day mortality were 0, respectively. Peri- and operative data are described in Table 3.

### 3.5. Follow Up and Overall Survival

The median follow-up of all patients with a combined resection of CRLM and vascular resection/reconstruction was 24.4 months (range 2.8–108.7). The median recurrence free survival (RFS) of all 32 patients was 6 months and the median overall survival (OS) was 33.3 months (COI 20.1–46.4 months). The 1-year survival rate was 80%.the 3-year survival rate 36% and the 5-year survival rate 18%. 

Dividing the cohort into two groups depending on a negative or positive histological vascular infiltration, the median survival in the group with vascular infiltration was 21 months and with no vascular infiltration 33.3 months and revealed no significant difference (*p* = 0.977). 

All patients with vascular infiltration have developed recurrent disease after a median of 3.5 months within the follow-up period (Table 4).

### 3.6. Case Report 1

The 50-year-old woman was referred to our hepatobiliary institution with synchronous bilateral CRLM in March 2011. The primary tumor (cancer of the colon sigmoideum) was resected in September 2010 at the initial point of diagnosis. She then received systemic chemotherapy with 5-FU, Oxaliplatin, Irinotecan and Bevacizumab for 3 months. The liver metastases involved the segments IVa/b, V, VII, VIII and I as well as II and III. Additionally, the CT-scan suspected an infiltration of the vena cava inferior (Figure 3). Based on the preoperative findings, an extended right hepatectomy with a resection/reconstruction of the VCI needed to be performed but the volume of segment II and III amounted only 315 mL. To avoid postoperative liver failure (PHLF) due to a too small future liver remnant (FLR) we planned an ALPPS procedure (associating liver partition and portal vein ligation for staged hepatectomy). 

During the first step, an infiltration of the VCI could not be excluded and therefore she received a partial resection of the VCI with direct suture as reconstruction (Figure 4a,b). The second step was completed on day 7 after the initial surgery and the FLR showed a volume increase of 102% (635 mL). The postoperative course was uneventful and she was discharged after 21 days. 

Histopathological examination showed no infiltration of the VCI and a R0 parenchymal resection margin was achieved. After 15.1 months she developed lung metastases which were treated with systemic chemotherapy. The patient died 50 months after the ALPPS procedure. 

### 3.7. Case Report 2

A 58-year-old man was admitted to our hospital in March 2013 with a recurrent single CRLM in segment VIII of the liver involving the VCI (Figure 5). The primary tumor was located in the right colon and has been resected in November 2009 (TNM-classification: pT4, pN2, M0, local R0). The patient received an atypical resection of segment VIII with a segmental resection of the VCI. Reconstruction was performed by interpositioning of a synthetic graft (Gore vascular graft^®^). The histopathological examination confirmed an infiltration of the vascular wall with tumor negative parenchymal and vascular resection margins (R0 parenchymal and vascular). No postoperative complications occurred and the patient was discharged on day 8 after combined liver and vena cava resection and reconstruction. Tumor recurrence occurred 15 months after liver surgery and involved the liver as well as abdominal lymph nodes. The patient received systemic chemotherapy over 12 months and was re-operated in March 2014 (R0 resection liver, lymph nodes showed complete remission under chemotherapy). Overall survival from the date of liver resection in combination with the VCI resection was 94 months. 

## 4. Discussion

Liver resection remains the standard of care in patients with colorectal liver metastases offering the best oncological outcome with 5-year survival rates up to 60%. Nevertheless, at the point of diagnosis only 15–30% of patients with CRLM are eligible for upfront surgery due to oncological, technical or functional reasons [18,19]. Particularly challenging are colorectal liver metastases with direct contact to major hepatic vascular structures (VCI, PV and HV) requiring complex surgical procedures, whereby the vena cava inferior and the hepatic veins are more frequently affected by CRLM than the portal vein [12,20]. 

Overall, liver surgery in combination with major vessel resection and reconstruction is a rare setting in the surgical management of CRLM. Analyzing our data of 1236 surgical liver approaches in patients with colorectal liver metastases, only 2.6% cases were combined liver and vascular resections due to a suspected tumor involvement of the above-mentioned major liver vessels. The small number of our cases (*n* = 36) is in accordance with the common literature and might be affiliated to the fact that colorectal liver metastases with suspected vascular tumor infiltration of the VCI, the PV or major HV are often declared to be a contraindication for a surgical approach being associated with a poor oncological prognosis and dismal surgical outcome [21,22]. 

This assessment was traditionally endorsed by the aim that a R0 parenchymal resection margin has to be the primary goal for resections of CRLM to reduce the risk of local intrahepatic recurrence. During the last decade, the impact of R0 parenchymal resection margins on the overall survival and oncological outcome has become less important due to the introduction of effective multimodal systemic treatment approaches as well as local treatment options such as as ablation, irreversible electroporation (IRE) or selective internal radiation therapy (SIRT). Thus, the definition of an adequate R0 surgical margin has changed from 1 cm to 1 mm and is nowadays widely accepted by oncologists and liver surgeons [23]. 

In addition to this, in patients with multiple and/or bilobar colorectal liver metastases a shift from extended anatomical resections towards parenchymal sparing surgical approaches appeared a relevant place in the surgical management to avoid postoperative liver dysfunction or liver failure and to increase the number of patients being eligible for liver resection. The achieved oncological long-term outcomes in cases of R0 parenchymal sparing situation did not differ significantly from extended anatomical resections but were associated with lower rates of morbidity and mortality [24,25,26]. 

Due to the improvements in operative techniques and surgical approaches in the setting of CRLM not only the number of patients being eligible for surgery increased but also the rate of R1 parenchymal resection margins. R1 parenchymal situations appear in 10–30% of liver resections being attributed to more aggressive surgical approaches for CRLM [27]. In our analysis of combined liver resections with vascular resection/reconstruction a R1 parenchymal margin occurred in 5/32 cases (15.6%) and confirms the above-mentioned data in the literature. 

In contrast to the mentioned data of the oncological outcome after R0 or R1 parenchymal resections, the role of the vascular resection margin in cases of CRLM with direct contact to major liver vessels still remains unclear. It has to be clarified whether tumor detachment from the related vessel is an adequate approach to remove the liver metastases to avoid a great loss of healthy liver parenchyma. Therefore, Torzilli et al. investigated the impact of the vascular resection margin of CRLM with the involvement of major vascular structures related to oncological outcome. They concluded that the oncological outcome after R1 vascular resections was not significantly worse than after R0 vascular resections, thus justifying the detachment of CRLM from intrahepatic vessels and facilitating parenchymal sparing surgery [28]. 

Nevertheless, data of liver resections in combination with vascular resection/reconstruction in the setting of CRLM are rare. Series with comparable liver resections with major vessel resections and reconstructions and a comparable number of patients to our study population report median OS rates ranging from 19 to 29 months [11,13,22,29,30]. Our median OS of 33.3 months is even slightly superior and supports that in cases of CRLM with major vessel involvement, a surgical approach as a chance for cure and best long-term survival seems to be justified. Even more, comparing the median overall survival of patients after surgery with the overall survival of patients in a palliative setting. 

The initially reported high rates of morbidity and mortality of combined extended liver resections and complex vascular reconstructions also hampered the implementation as a standard surgical procedure in the treatment of colorectal liver metastases. But the shift in surgical techniques away from major resections with total vascular occlusion or even ante-situm/in-situm or ex-situm procedures with hypothermic perfusion and towards parenchymal sparing procedures and less “invasive” vascular techniques such as selective clamping of the affected vessel whenever possible lead to low and acceptable morbidity and mortality rates. In or study group no in-hospital or 90-day mortality occurred and no severe morbidity, meaning postoperative liver failure or life-threatening complications were observed [11,13,31,32,33,34,35]. 

The acceptance of R1 vascular margin situations in advanced liver resection with vascular resection and reconstruction might be supported by the fact that in our study population a histological proven vessel involvement occurred in only six cases. The result emphasizes that this scenario seems to be a rare event. 

Nevertheless, since even at the point of operation a vascular invasion cannot be predicted reliably our preferred institutional strategy in patients with CRLM and suspected major vessel involvement is still to perform vascular resection and reconstruction to achieve an R0 resection.

We follow this concept since it is in accordance with basic principles of oncologic surgery, at least theoretically offering a somewhat higher chance for cure but at the same time not burdened by additional morbidity or mortality.

Our data confirm that even advanced hepatic surgery combined with vascular resection/reconstruction can be performed with low rates of morbidity and no mortality independent of the involved vascular structure (VCI/PV/HV) and type of reconstruction (primary suture/patch plastic/synthetic graft replacement). These results correspond to previously reported data on morbidity and mortality in the literature [22,33]. 

The study has some limitations. First, the number of cases (*n* = 32) of the presented study is too small for valid conclusions or to even prove an oncologic benefit resulting from this approach, as more data for long term analysis are lacking. Second, the study is a retrospective analysis and might implicate a bias in patient selection. 

Nevertheless, our analysis includes one of the highest number of cases with vascular resection for CRLM published so far [13,22,29,36]. 

## 5. Conclusions

In conclusion, liver resection for colorectal liver metastases with simultaneous vascular resection/reconstruction is rare but can be performed with low morbidity rates. Furthermore, tumor infiltration is occurs seldom and its impact on overall survival is unclear. Our preferred intention-to-treat concept in cases of CRLM with suspected involvement of large hepatic vessels is parenchymal sparing liver resection with vascular resection and reconstruction to achieve a negative margin. 

In technically difficult cases with a supposed higher risk for postoperative complications and since histological vascular infiltration remains a rare setting, tumor detachment from vessels without vascular resection and achievement of a R1 vascular situation seems to be a most reasonable surgical alternative.

## Figures and Tables

**Figure 1 cancers-16-00571-f001:**
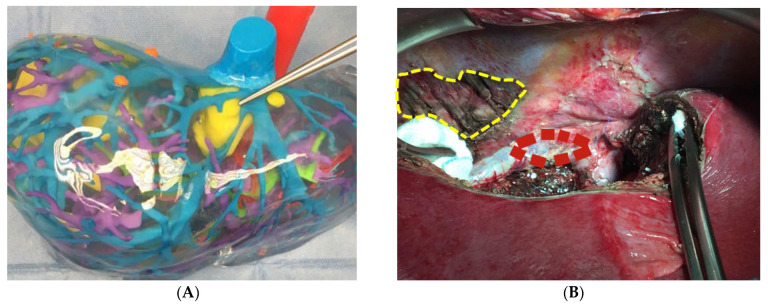
D-Reconstruction of the liver (**A**) and reconstruction of the right hepatic vein by using a peritoneal patch (**B**).

**Figure 2 cancers-16-00571-f002:**
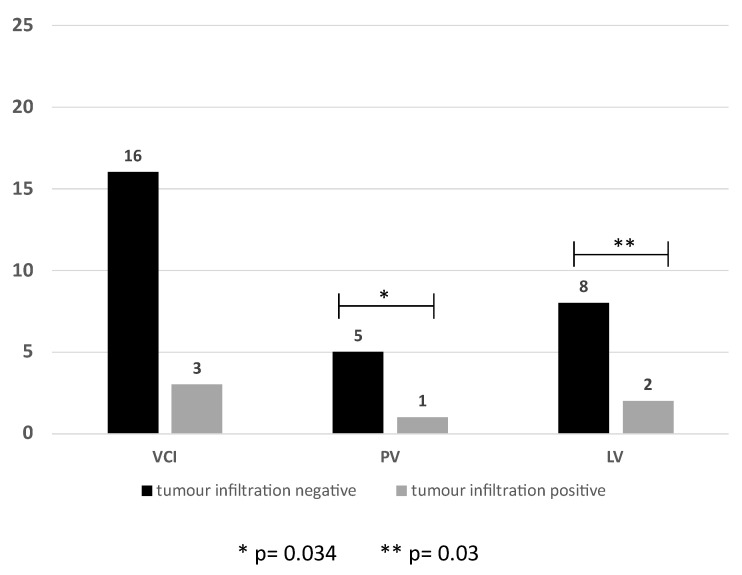
Ratio of positive and negative histopathological tumor infiltration of the VCI, PV and LV, VCI: vena cava inferior PV: portal vein LV: liver vein.

**Figure 3 cancers-16-00571-f003:**
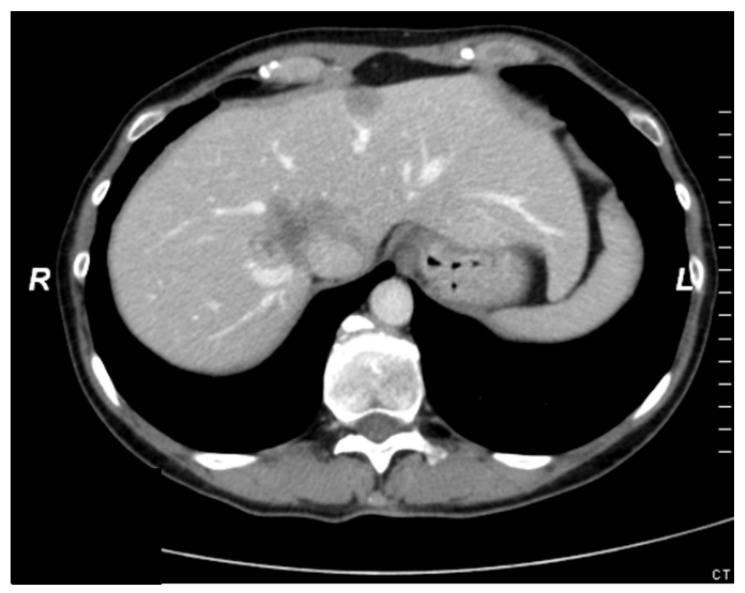
Preoperative CT-scan with a CRLM involving the VCI (Case report 1).

**Figure 4 cancers-16-00571-f004:**
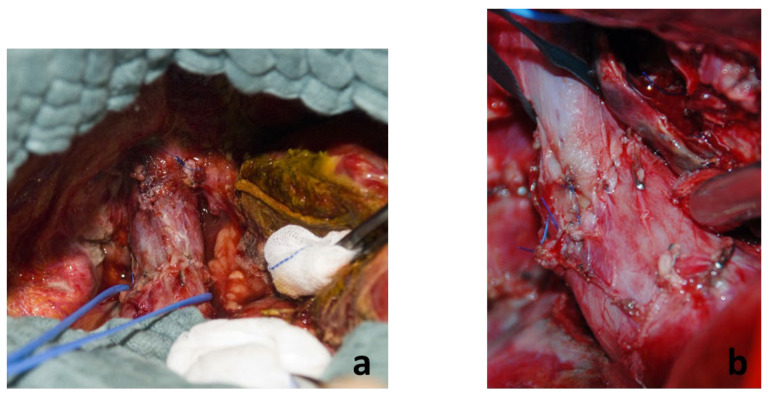
(**a**,**b**) Reconstruction vena cava inferior with direct suture.

**Figure 5 cancers-16-00571-f005:**
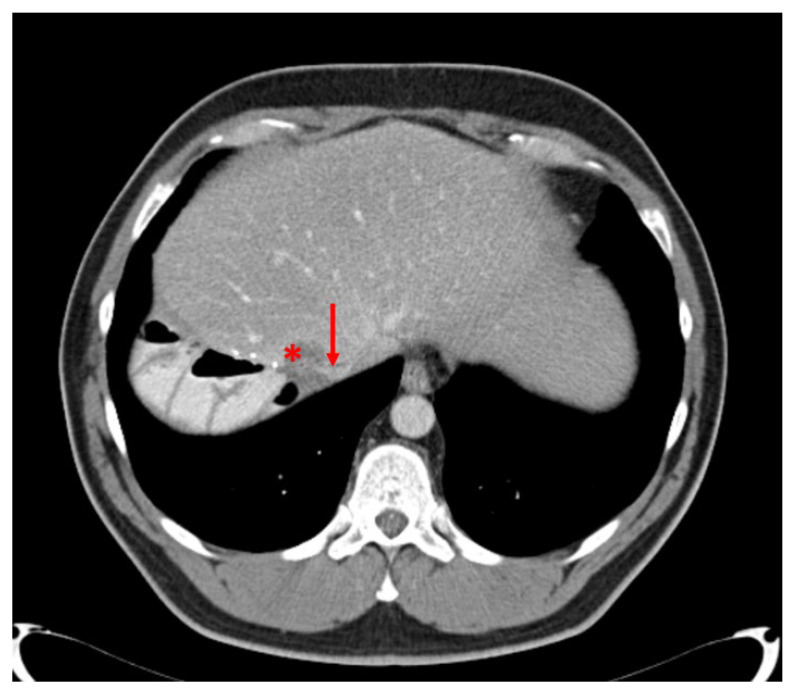
Preoperative CT scan (Case report 2); * colorectal liver metastases; ↓ vena cava inferior.

**Table 1 cancers-16-00571-t001:** Type of vascular reconstructions.

	Direct Suture	Peritoneal Patch	Bovine Patch	Gore Vascular Graft
Vena cava inferior; *n* = 19	11	1	6	1
Portal vein; *n* = 6	6	0	0	0
Hepatic vein *n* = 10	6	3	1	0

**Table 2 cancers-16-00571-t002:** Patients’ demographics.

Parameter	Negative Vascular Infiltration (*n* = 26)	Positive Vascular Infiltration (*n* =6)	*p* Value
Age (yr, median, range)	59 (48–72)	64 (50–76)	0.39
Gender			
Male	14	5	0.19
Female	12	1	
ASA classification (median, range)	2.5 (2–3)	2 (2–3)	0.14
Primary tumour site			
Colon	12	4	0.37
Rectum	14	2	
Primary tumour nodal status			
Positive	18	4	
Negative	6	2	0.72
NA *	2	-	
Liver metastases			
Synchronous	20	2	0.04
Metachronous	6	4	
Number of liver metastases (median, range)	2 (1–11)	2 (1–3)	0.96
Size of liver metastases (median, range)	3.9 (0.5–9)	4.5 (2–10.5)	0.53
Extend of liver metastases			
Solitary	6	2	0.60
Multifocal	20	4	
Unilateral	9	4	0.15
Bilateral	17	2	
Recurrent liver metastases			
Yes	12	3	0.87
No	14	3	
Preoperative chemotherapy			
Yes	16	2	0.21
No	10	4	

NA *: not available.

**Table 3 cancers-16-00571-t003:** Operative data and perioperative details.

Parameter	Negative Vascular Infiltration (*n* = 26)	Positive Vascular Infiltration (*n* = 6)	*p* Value
Extend of hepatectomy			
Minor	11	3	0.92
Major	11	2	
Extended resection	4	1	
One staged hepatectomy	24	6	0.78
Two staged hepatectomy	1	0	
ALPPS	1	0	
Resected vessels *			
Portal vein	4	2	0.31
Inferior vena cava	15	4	0.16
Hepatic vein	8	2	0.90
Reconstruction			
Primary suture	18	5	0.09
Patch plastic	9	2	
Synthetic graft replacement	-	1	
Resection margin			
R0	24	3	0.01
R1	2	3	
Hospital stay (days, median, range)	19 (7–40)	12 (7–31)	0.48
Morbidity (≥ grade 3 **)			
Yes	10	3	0.60
No	16	3	
90-day mortality	0	0	

* Group without vascular infiltration: 1 patient received combined resection of VCI and LV. ** Dindo-Clavien classification.

**Table 4 cancers-16-00571-t004:** Demographics and outcome of patients with vascular infiltration of a major vein (V2).

Patient	Gender	Age	Primary Tumour Site	Liver Metastases	Resected Vessel	Reconstruction	Extend of Liver Resection	Resection Margin	Number of Metastases	Hospital Stay (Days)	Morbidity (≥Grade 3)	Recurrence (Yes/No)	Recurrence Site	Follow Up/Status
1	M	58	Colon	metachronous	VCI	Synthetic graft replacement	minor	R0	1	8	no	yes	disseminated	died after 93.83 months
2	F	56	Colon	synchronous	VCI + PV	Primary suture	minor	R0	2	10	no	yes	disseminated	died after 7.93 months
3	M	74	Colon	metachronous	PV	Primary suture	major	R0	>5	14	3a	yes	lung	alive after 40.27 months
4	M	76	Rectum	metachronous	HV	Patch plastic	minor	R1	3	7	no	yes	liver	alive after 10.33 months
5	M	50	Colon	synchronous	VCI	Patch plastic	major	R1	2	31	3a	yes	liver	alive 10.47 months
6	M	70	Rectum	metachronous	VCI + HV	Primary suture	extended	R1	1	15	3a	no	-	alive after 3.9 months

M: Male; F: Female; VCI: Vena cava inferior; PV: Portal vein; HV: Hepatic vein.

## Data Availability

The data presented in this study are available in this article.

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
