# Peer review of "Resection of Colorectal Liver Metastases with Major Vessel Involvement"

_cancers, 2024, doi:10.3390/cancers16030571_

Round 1
Reviewer 1 Report (Previous Reviewer 1)
Comments and Suggestions for Authors
The revised paper is improved. However, since the case series began in 2008 I think the author should be able to provide a five-year overall survival rate besides the one and three year that is presented. Even if this is somewhat low overall survival at five years, it’s important for the readership to know this.
Author Response
Dear Reviewer 1,
thank you for your improvement suggestions. We therefore added the 5-year-survival rate of 18% (page 8 line 21).
We appreciate your support.
Best regards.
Janine Baumgart
Reviewer 2 Report (Previous Reviewer 2)
Comments and Suggestions for Authors
The overall weakness of the manuscript is the small patient number which precludes any meaningful analysis.
Author Response
Dear Reviewer 2,
thank you for your comment on our manuscript. Nevertheless, regarding the published literature, there are few equivalent and no higher numbers of patients which received a liver resection in combination with major vessel resection/ reconstruction due to colorectal liver metastases as far as we know.
Certainly, the small number of patients causes limitations for valid conclusions. Therefore, we named this point of criticism in our discussion (page 14, line 13) .
Thank you for your remarks and improvement suggestions.
Best regards,
Janine Baumgart
Round 2
Reviewer 2 Report (Previous Reviewer 2)
Comments and Suggestions for Authors
I still hold my opinion that the patient number is small in this study, that precludes any meaningful analysis. If the journal accepts case series, I am fine with that.
Comments on the Quality of English LanguageNA
This manuscript is a resubmission of an earlier submission. The following is a list of the peer review reports and author responses from that submission.
Round 1
Reviewer 1 Report
Comments and Suggestions for Authors
The authors present a small series of 32 patients with CRLM who underwent major hepatic vessel resection and reconstruction. vascular infiltration was shown in 6/32 patients which had a trend towards worse median OS. This is a small series so it is hard to draw meaningful conclusions. The authors should provide 1,3,5-yr RFS and OS in the two groups, and compare to their contemporaneous CRLM resections without vascular resections.
Reviewer 2 Report
Comments and Suggestions for Authors
Overall, this manuscript carries low value because of the small patient number and difficulty to generalise the clinical massage. I agree with the previous reviewer that only 6 out of 32 patients had histological proof of vascular involvement by tumor. In other words, the rest of patients may not need major vascular reconstruction because of absence of tumor invasion to major vessels.
Comments on the Quality of English LanguageNil
